# Energetics of Baird aromaticity supported by inversion of photoexcited chiral [4n]annulene derivatives

Michihisa Ueda[1], Kjell Jorner [2], Young Mo Sung[3], Tadashi Mori [4], Qi Xiao[1], Dongho Kim[3], Henrik Ottosson[2], Takuzo Aida [1,5] & Yoshimitsu Itoh [1]

For the concept of aromaticity, energetic quantification is crucial. However, this has been elusive for excited-state (Baird) aromaticity. Here we report our serendipitous discovery of two nonplanar thiophene-fused chiral [4n]annulenes **Th4COT_Saddle** and **Th6CDH_Screw**, which by computational analysis turned out to be a pair of molecules suitable for energetic quantification of Baird aromaticity. Their enantiomers were separable chromatographically but racemized thermally, enabling investigation of the ring inversion kinetics. In contrast to **Th6CDH_Screw**, which inverts through a nonplanar transition state, the inversion of **Th4COT_Saddle**, progressing through a planar transition state, was remarkably accelerated upon photoexcitation. As predicted by Baird's theory, the planar conformation of **Th4COT_Saddle** is stabilized in the photoexcited state, thereby enabling lower activation enthalpy than that in the ground state. The lowering of the activation enthalpy, i.e., the energetic impact of excited-state aromaticity, was quantified experimentally to be as high as 21–22 kcal mol$^{-1}$.

[1] Department of Chemistry and Biotechnology, School of Engineering, The University of Tokyo, 7-3-1 Hongo, Bunkyo-ku, Tokyo 113-8656, Japan. [2] Department of Chemistry – Ångström Laboratory, Uppsala University, Box 523, Uppsala 751 20, Sweden. [3] Spectroscopy Laboratory for Functional $\pi$-Electronic Systems and Department of Chemistry, Yonsei University, Seoul 120-749, Korea. [4] Department of Applied Chemistry, Graduate School of Engineering, Osaka University, 2-1 Yamada-oka, Suita, Osaka 565-0871, Japan. [5] RIKEN Center for Emergent Matter Science, 2-1 Hirosawa, Wako, Saitama 351-0198, Japan. Correspondence and requests for materials should be addressed to D.K. (email: dongho@yonsei.ac.kr) or to H.O. (email: henrik.ottosson@kemi.uu.se) or to T.A. (email: aida@macro.t.u-tokyo.ac.jp) or to Y.I. (email: itoh@macro.t.u-tokyo.ac.jp)

nnulenes are monocyclic hydrocarbons comprising alternating single and double bonds, whose preferred conformations in the electronic ground ($S_0$) state are determined by the number of their $\pi$-electrons[1]. Annulenes with $4n + 2$ $\pi$-electrons are categorized as Hückel aromatic compounds that prefer to adopt a bond-length equalized planar conformation because the electronic conjugation enabled by planarization leads to energetic stabilization[2, 3]. In contrast, annulenes with $4n$ $\pi$-electrons are categorized as Hückel antiaromatic compounds that tend to adopt a bond-length alternate nonplanar conformation because of their unfavorable electronic conjugation and/or increased ring strain[4–8]. In 1972, Baird theoretically deduced[9] that [$4n$]annulenes in the triplet excited ($T_1$) state, just like [$4n + 2$]annulenes in the $S_0$ state, prefer to adopt a planar conformation because the resulting electronic conjugation leads to energetic stabilization[10, 11]. Later, Baird's theory was shown computationally to be also applicable to the singlet excited ($S_1$) state[12–14]. Indeed, a series of photochemical and photophysical phenomena that cannot be explained by Hückel's rule have been reasonably explained by Baird's rule[15, 16]. For example, Wan et al. utilized the concept of excited-state aromaticity to explain why certain $S_N1$-type substitution reactions involving cyclic intermediates with $4n$ $\pi$-electrons are facilitated by photo-irradiation[17–20]. Ottosson, Kilså, and coworkers[21, 22] showed that

Baird's rule can account for how the excitation energy of fulvene changes with its substituents. Indeed, Ottosson, and coworkers[23] postulated that Baird's theory is a useful back-of-an-envelope tool for the design and exploration of novel photofunctional materials, and also showed that it can be used to develop new photoreactions[24, 25]. As a seminal achievement to elucidate Baird's rule, Kim, Osuka, and coworkers recently provided spectroscopic evidence of Baird aromaticity based on the transient absorption spectral profiles of a particular type of expanded porphyrin derivative[26–28]. Equally important for experimentally substantiating Baird's rule is to energetically quantify the concept of excited-state aromaticity[6]. However, this essential issue has not been addressed because annulene derivatives that fulfill certain prerequisites are currently unavailable.

In the course of our study on the synthesis of thiophene oligomers using a modified Ullmann coupling reaction, we noticed two different chiral [$4n$]annulene derivatives as by-products: **Th4COT**$_{Saddle}$ and **Th6CDH**$_{Screw}$. Single-crystal X-ray crystallographic analysis revealed that these by-products are nonplanar and conformationally chiral. We succeeded in their optical resolution using high-performance liquid chromatography on a chiral stationary phase (chiral HPLC). Circular dichroism (CD) spectroscopy in methylcyclohexane showed that the enantiomers of **Th4COT**$_{Saddle}$ and **Th6CDH**$_{Screw}$ are racemized

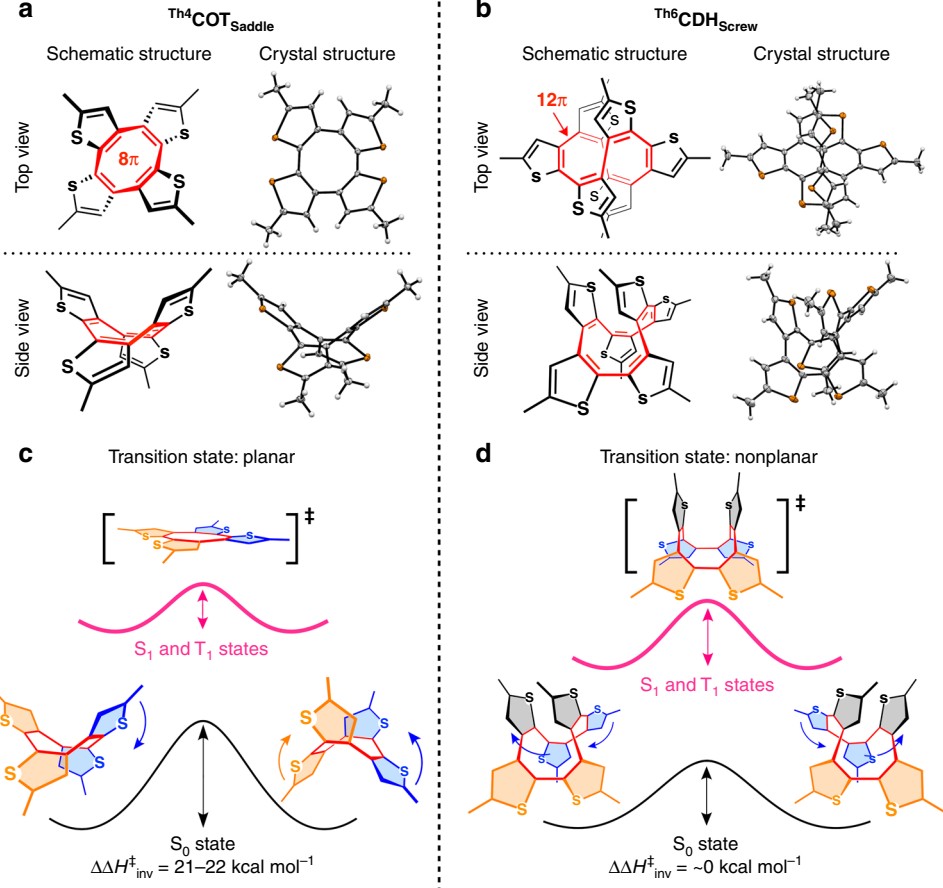

**Fig. 1** [$4n$]Annulene derivatives with and without Baird aromaticity upon photoexcitation. **a**, **b**, Molecular structures and ORTEP drawings (50% ellipsoid probability) of **Th4COT**$_{Saddle}$ (**a**) and **Th6CDH**$_{Screw}$ (**b**), which have 8$\pi$-electron and 12$\pi$-electron annulene cores (*red colored*), respectively. **c**, **d**, Schematic illustrations of the energy barriers for the ring inversion events of **Th4COT**$_{Saddle}$ (**c**) and **Th6CDH**$_{Screw}$ (**d**), where *colored arrows* represent the movement directions of the thiophene rings of the same color. **Th4COT**$_{Saddle}$ and **Th6CDH**$_{Screw}$ invert through planar and nonplanar transition states, respectively, as shown in the square brackets. *Black* and *pink-colored solid curves* represent energy barriers in the ground and photoexcited states, respectively. Upon photoexcitation, the activation enthalpy for the ring inversion ($\Delta H^{\ddagger}_{inv}$) of **Th4COT**$_{Saddle}$ is lowered by 21–22 kcal mol$^{-1}$, but that of **Th6CDH**$_{Screw}$ remains unchanged

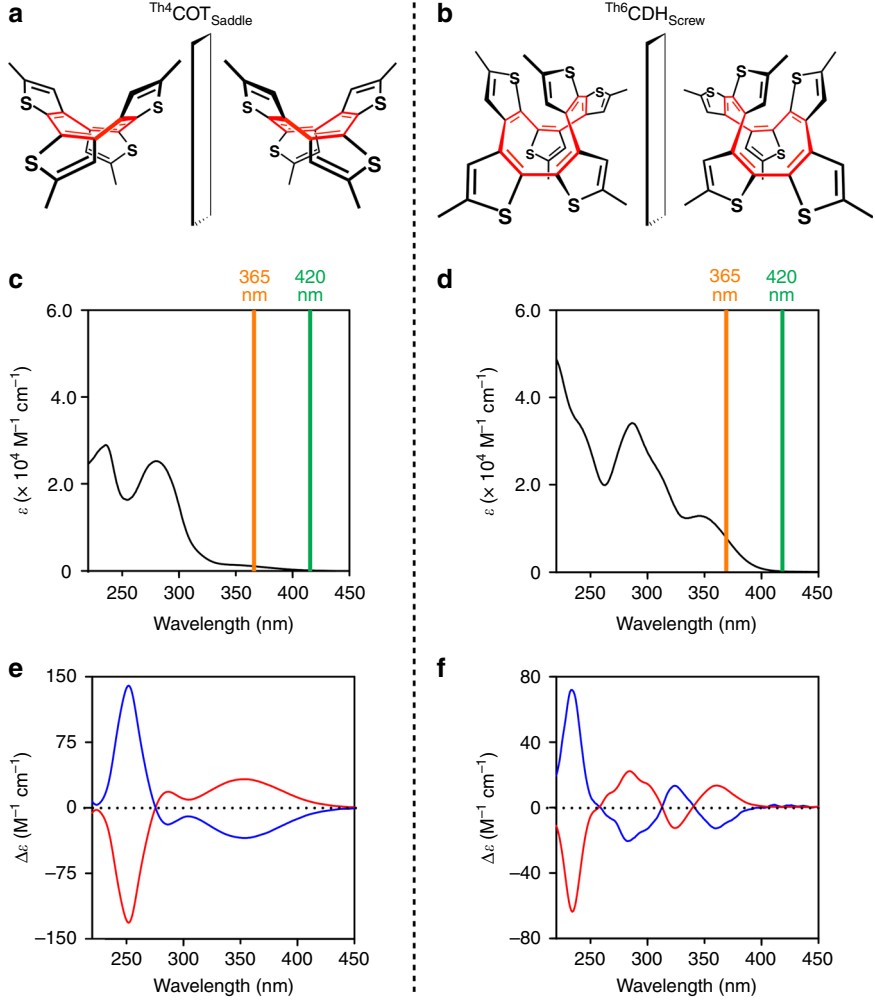

**Fig. 2** Optical features of $^{Th4}COT_{Saddle}$ and $^{Th6}CDH_{Screw}$. **a**, **b**, Schematic structures of the enantiomers of $^{Th4}COT_{Saddle}$ (**a**) and $^{Th6}CDH_{Screw}$ (**b**). **c**, **d**, Electronic absorption spectra of $^{Th4}COT_{Saddle}$ (**c**) and $^{Th6}CDH_{Screw}$ (**d**) in methylcyclohexane at 25 °C. *Orange-* and *green-colored vertical lines* correspond to the wavelengths of $\lambda = 365$ and 420 nm, which were utilized for direct and sensitized photoexcitation of the molecules, respectively. **e**, **f**, Circular dichroism (CD) spectra of $^{Th4}COT_{Saddle}$ (**e**) and $^{Th6}CDH_{Screw}$ (**f**) in methylcyclohexane at 25 and –20 °C, respectively. *Blue-* and *red-colored traces* represent the spectra of compounds obtained as the first and second fractions of the chiral HPLC, respectively

through ring inversion under appropriate conditions. Density functional theory (DFT) calculations indicated that $^{Th4}COT_{Saddle}$ adopts a planar conformation, whereas $^{Th6}CDH_{Screw}$ adopts a nonplanar conformation, in the transition state of their ring inversion processes. Of particular importance, we found that the racemization of $^{Th4}COT_{Saddle}$ is remarkably accelerated by its photoexcitation, whereas that of $^{Th6}CDH_{Screw}$ is unaffected by photoexcitation. We wondered whether these contrasting results are related to a prime issue of excited-state aromaticity, i.e., Baird aromaticity. In fact, quantum chemical computational analysis revealed that $^{Th4}COT_{Saddle}$ and $^{Th6}CDH_{Screw}$ are a pair of molecules suitable for energetic quantification of Baird aromaticity. Previous studies demonstrated experimentally that cyclooctatetraene (COT), oxepin, and thiepin analogs in the photoexcited state adopt planar conformations[29, 30]. However, they are planarized only in a barrierless manner, precluding energetic quantification of photoexcited planar [$4n$]annulenes. On the other hand, $^{Th4}COT_{Saddle}$ in the present study provides a sterically congested ring inversion process with a positive activation enthalpy. Because the transition state of the ring inversion of $^{Th4}COT_{Saddle}$ involves its planarized core, the kinetic studies both in the $S_0$ state and photoexcited ($S_1/T_1$) state surely enable us to support Baird's rule from an energetic viewpoint.

## Results

**Synthesis and optical resolution of $^{Th4}COT_{Saddle}$ and $^{Th6}CDH_{Screw}$.** A $\beta$-linked thiophene dimer (**Th2**) was subjected to a modified Ullmann coupling reaction, and the crude reaction mixture was passed through a silica gel short column and submitted to size exclusion chromatography on a polystyrene gel column for the isolation of $^{Th4}COT_{Saddle}$ and $^{Th6}CDH_{Screw}$ (Supplementary Methods). $^{Th4}COT_{Saddle}$ carries a COT ([8]annulene) core[31, 32], while $^{Th6}CDH_{Screw}$ bears a cyclododecahexaene (CDH: [12]annulene) core[33]. Through vapor diffusion, all these annulenes afforded single crystals suitable for X-ray crystallography (Fig. 1a, b). The crystal structure of $^{Th4}COT_{Saddle}$ adopts a highly nonplanar conformation in its annulene core with a bond-length alternation. A similar structural feature was observed for the crystal of $^{Th6}CDH_{Screw}$. Chiral HPLC (Supplementary Methods) was used to separate these chiral compounds into the corresponding enantiomers (Fig. 2e, f). Their enantiomers were thermally racemizable, which encouraged us to investigate the ring inversion kinetics. The enantiomers of $^{Th4}COT_{Saddle}$, when heated above 40 °C in methylcyclohexane, underwent racemization (Fig. 3a), while those of $^{Th6}CDH_{Screw}$ were much more prone to racemization; the enantiomers of $^{Th6}CDH_{Screw}$ were racemized unless the compound was cooled below –40 °C (Fig. 3b).

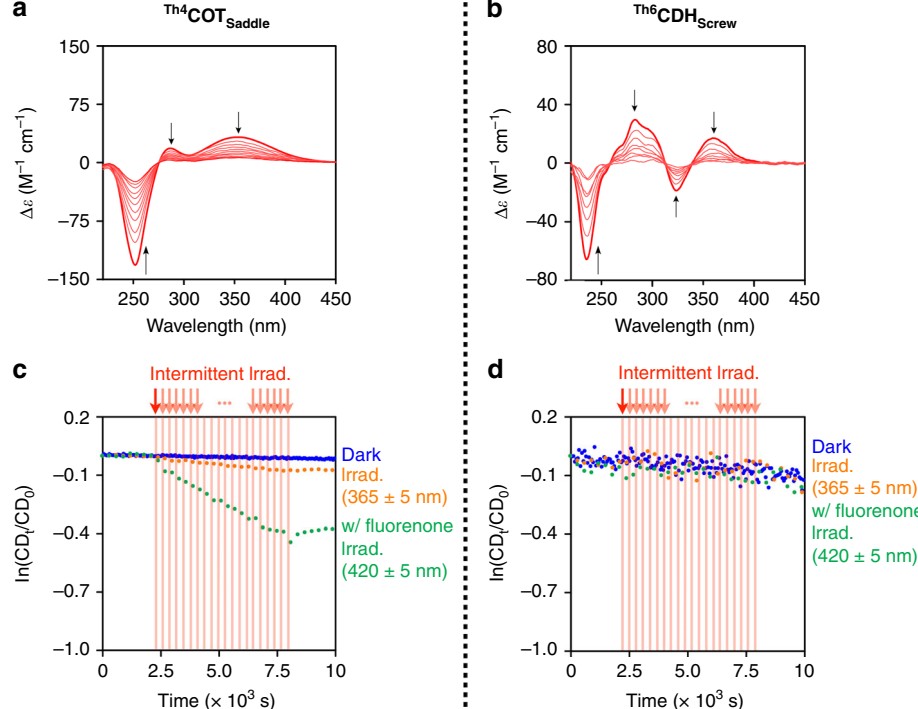

**Fig. 3** Ring inversion kinetics of **Th4COT_Saddle** and **Th6CDH_Screw**. **a**, **b**, Time-dependent CD spectral change profiles (240-s interval) of **Th4COT_Saddle** (**a**) and **Th6CDH_Screw** (**b**) in methylcyclohexane at 60 and 0 °C, respectively. *Black arrows* in **a** and **b** represent the directions of the CD spectral change with time. **c**, **d**, Decay profiles of the CD intensities at 260 nm (**c**, **Th4COT_Saddle**) and 280 nm (**d**, **Th6CDH_Screw**) in deaerated methylcyclohexane at 20 and −20 °C, respectively. *Blue-colored dots* represent decay profiles without photoirradiation. *Orange-colored dots* represent decay profiles with photoirradiation ($\lambda = 365 \pm 5$ nm). *Green-colored dots* represent decay profiles with photoirradiation ($\lambda = 420 \pm 5$ nm) in the presence of fluorenone (0.34 equiv. for **Th4COT_Saddle** and 0.23 equiv. for **Th6CDH_Screw**). Photoirradiation (250 s, *red vertical lines*) and CD spectroscopy (50 s, *white area between red vertical lines*) were conducted alternately for 6000 s

**Ring inversion in the electronic ground and excited states**. The inversion kinetics of **Th4COT_Saddle** and **Th6CDH_Screw** were studied in deaerated methylcyclohexane. Neither compound decomposed under photoexcitation (Supplementary Figs. 8 and 9). For the selective generation of the $S_1$ state, **Th4COT_Saddle** and **Th6CDH_Screw** were photoexcited at their longest wavelength absorption bands ($\lambda = 365 \pm 5$ nm, Fig. 2c, d). At 20 °C in the dark, the CD intensity of **Th4COT_Saddle**, as described above, remained unchanged over a period of 3 h (Fig. 3c, *blue dots*). However, when the solution was irradiated under otherwise identical conditions, the CD intensity gradually decreased (half-life ($t_{1/2}$) = 12 h) (Fig. 3c, *orange dots*). When the photoirradiation was stopped, the CD intensity no longer decreased. Transient absorption spectroscopy (TAS) of **Th4COT_Saddle** upon photoexcitation at $\lambda = 355$ nm enabled us to detect the $S_1$ state dynamics, which decayed exponentially with a lifetime of 5.5 ps (Supplementary Fig. 16). No other excited species, including the $T_1$ state, were detected. Hence, we conclude that the $S_1$ state of **Th4COT_Saddle** can facilitate its ring inversion.

Because Baird aromaticity has typically been discussed in the $T_1$ state[9, 11, 34–36], it is important to confirm whether the ring inversion of **Th4COT_Saddle** is facilitated in its triplet excited state. Therefore, we selectively generated the $T_1$ state of **Th4COT_Saddle** by photoexcitation of fluorenone (0.34 equiv., $E_T = 50.9$ kcal mol$^{-1}$, $\varphi_T = 1.00$[37]) at $\lambda = 420 \pm 5$ nm as a triplet photosensitizer and investigated the ring inversion process of **Th4COT_Saddle** at 20 °C. Although **Th4COT_Saddle** does not absorb light in this wavelength region ($\varepsilon = \sim 60$ cm$^{-1}$ M$^{-1}$; Fig. 2c), the ring inversion of **Th4COT_Saddle** was remarkably facilitated ($t_{1/2} = 2.1$ h) by the photoexcitation of fluorenone (Fig. 3c, *green dots*, *red-striped zone*). Without photoirradiation, fluorenone itself did not

facilitate the ring inversion (Fig. 3c, *green dots*, before and after *red-striped zone*). The triplet sensitization should occur smoothly, considering that the $T_1$ state energy of **Th4COT_Saddle** is supposedly slightly lower than that of fluorenone (Supplementary Discussion). In fact, a photoexcited $T_1$ species of **Th4COT_Saddle** with a lifetime of 2 μs was observed by TAS upon photoexcitation at $\lambda = 420$ nm in the presence of fluorenone (Supplementary Fig. 23). Just in case, when the system was bubbled with oxygen, a possible triplet quencher[38], ring inversion was no longer facilitated by photoexcitation (Supplementary Fig. 11). These observations strongly support that the photo-accelerated ring inversion of **Th4COT_Saddle** in the presence of fluorenone originates from its $T_1$ state. As expected, the ring inversion of **Th4COT_Saddle** in the $S_1$ state was unaffected by $O_2$ (Supplementary Fig. 10).

Baird's rule explains why [4$n$]annulenes prefer to be planarized in their photoexcited states. As described in Fig. 4a, the ring inversion of **Th4COT_Saddle** involves its planar transition state. In contrast, **Th6CDH_Screw** cannot be planarized during its ring inversion process (Fig. 4b). Does photoexcitation affect the ring inversion of **Th6CDH_Screw**? Compared with **Th4COT_Saddle**, **Th6CDH_Screw** is more prone to thermal ring inversion in the dark, and its enantiomers were racemized in methylcyclohexane, even at −20 °C (Fig. 3d, *blue dots*, $t_{1/2} = 18$ h). Initially, a methylcyclohexane solution of **Th6CDH_Screw** at −20 °C was photoexcited at the longest wavelength absorption band ($\lambda = 365 \pm 5$ nm). However, no acceleration was observed for its ring inversion. TAS of **Th6CDH_Screw** with photoexcitation at $\lambda = 355$ nm indicated the presence of both $S_1$ and $T_1$ states with lifetimes of 70 ps and 350 ns, respectively (Supplementary Figs. 18 and 19). No accelerated ring inversion occurred

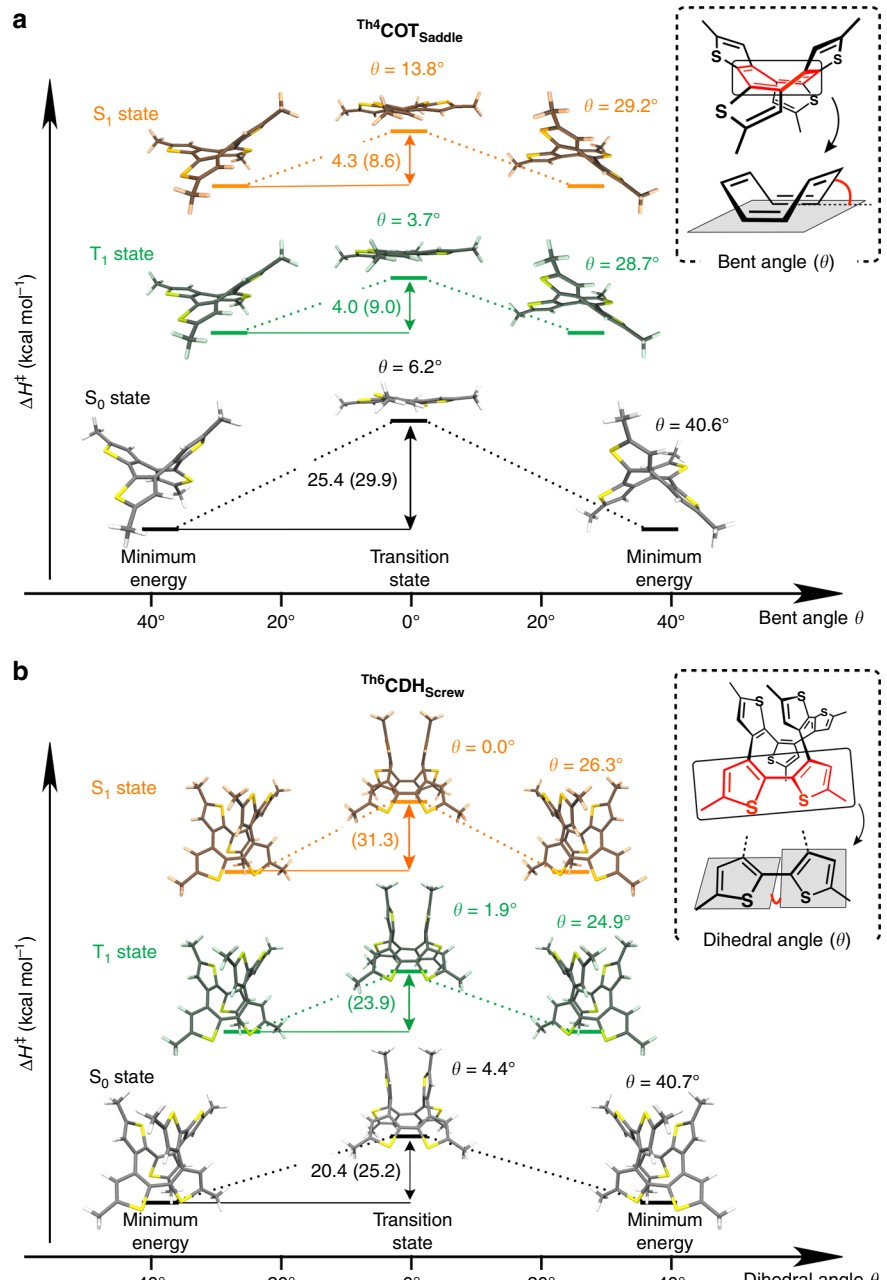

**Fig. 4** Ring inversion energetics of **Th4COT_Saddle** and **Th6CDH_Screw**. **a**, Conformational change of **Th4COT_Saddle** calculated along the reaction coordinate. **b**, Conformational change of **Th6CDH_Screw** calculated along the reaction coordinate. *Gray*-, *green*- and *orange*-colored drawings represent the conformations in the $S_0$, $T_1$, and $S_1$ states, respectively. For the $S_0$ and $T_1$ states, the structures were optimized at the B3LYP-D3(BJ)/6-31G(d) level, and the energy values were calculated by single-point calculations at the B3LYP-D3(BJ)/6-311+G(d,p) level using the optimized geometry. For the $S_1$ state, the structures were optimized at the TD-B3LYP-D3(BJ)/6-31+G(d,p) level, and the energy values were obtained by the same level of calculation used for the structural optimization. Activation enthalpies experimentally obtained (computationally calculated) are given in kcal mol$^{-1}$

when **Th6CDH_Screw** was placed under triplet sensitization conditions using photoexcited fluorenone (0.23 equiv.) at $\lambda = 420 \pm 5$ nm. The contrasting results with planarizable **Th4COT_Saddle** and non-planarizable **Th6CDH_Screw** appear reasonable, if the observed phenomena are dominated by Baird's rule. The possible effect of local heating (photothermal activation) on the accelerated ring inversion of **Th4COT_Saddle** was excluded, as described in the Supplementary Discussion. All these observations indicate that kinetic analysis of the ring inversion events of photoexcited- and ground-state **Th4COT_Saddle**, when compared to those of **Th6CDH_Screw**, enable energetic quantification of Baird aromaticity.

**Experimental evaluation of the activation enthalpies of ring inversion**. We investigated the ring inversion processes of **Th4COT_Saddle** and **Th6CDH_Screw** at varying temperatures in methylcyclohexane and analyzed their kinetic profiles using the Eyring equation (Supplementary Discussion). Based on the CD spectral decay profiles of **Th4COT_Saddle** at 40, 50, and 60 °C in the dark, the activation enthalpy of its ring inversion in the ground ($S_0$) state was evaluated as 25.4 kcal mol$^{-1}$ (Supplementary Fig. 13a, c). The decay profiles of **Th4COT_Saddle** at 0, 10, and 20 °C upon photoexcitation afforded activation enthalpies in the $S_1$ and $T_1$ states of 4.3 and 4.0 kcal mol$^{-1}$, respectively (Supplementary Figs. 14 and 15). The calculated activation enthalpies at (TD-)

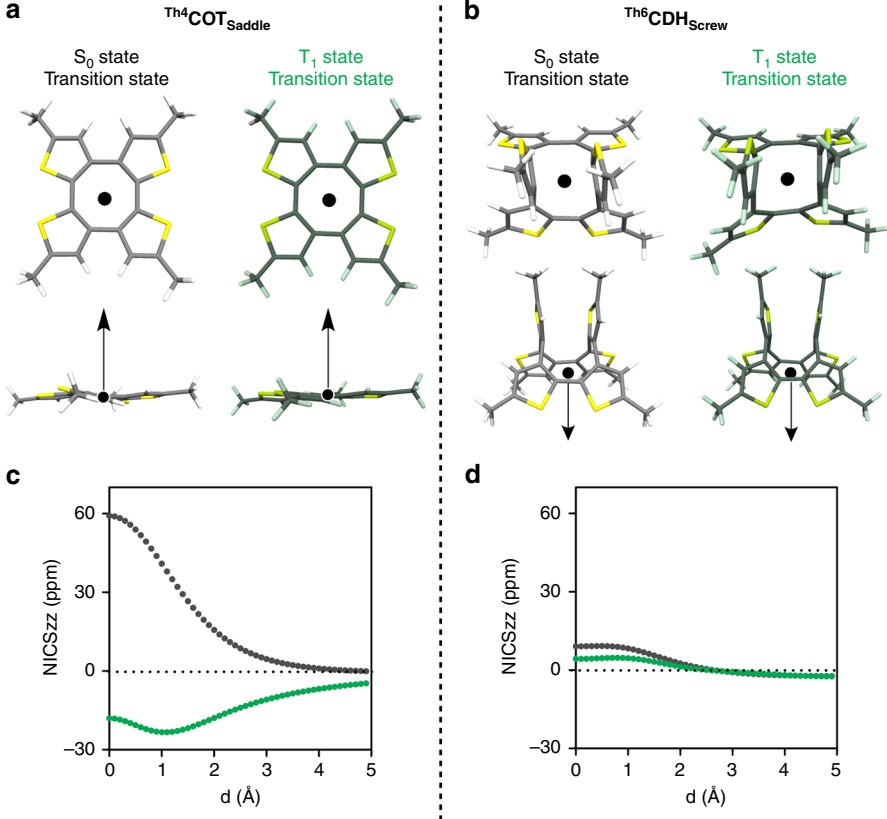

**Fig. 5** Nucleus-independent chemical shift (NICS) scans of $^{Th4}COT_{Saddle}$ and $^{Th6}CDH_{Screw}$. **a**, **b**, *Top* and *side* views of the inversion transition states of $^{Th4}COT_{Saddle}$ (**a**) and $^{Th6}CDH_{Screw}$ (**b**) in the $S_0$ and $T_1$ states. **c**, **d**, NICS$_{zz}$ scans of $^{Th4}COT_{Saddle}$ (**c**) and $^{Th6}CDH_{Screw}$ (**d**), which start from the annulene ring center and scanned along the *arrows* shown in **a** and **b**. *Black-* and *green-colored dots* represent the scans in the $S_0$ and $T_1$ states, respectively

B3LYP-D3(BJ) were 29.9, 8.6, and 9.0 kcal mol$^{-1}$ for the $S_0$, $S_1$, and $T_1$ states, respectively, which are in agreement with the experimental values. It is now clear that the planar transition state in the ring inversion of $^{Th4}COT_{Saddle}$ is photochemically stabilized in the excited ($S_1/T_1$) states. In sharp contrast, as described in Fig. 3d, the inversion rate of $^{Th6}CDH_{Screw}$ is unaffected by photoexcitation; thus, the activation enthalpies in the photoexcited ($S_1/T_1$) states would not differ from that in the $S_0$ state (20.4 kcal mol$^{-1}$, Supplementary Fig. 13b, d). In other words, no photochemical stabilization occurs at the nonplanar transition state in the ring inversion of $^{Th6}CDH_{Screw}$. This is supported by the calculated values at (TD-)B3LYP-D3(BJ) of 25.2, 31.3, and 23.9 kcal mol$^{-1}$ for the $S_0$, $S_1$, and $T_1$ states, respectively. The energetic quantification results are consistent with Baird's rule.

**Computational investigation of energetics and aromaticity**. Quantum chemical calculations are essential to clarify whether our observations, involving photochemical planarization of [4n]annulenes, is indeed caused by Baird aromaticity. In [4n] annulenes with fused arene moieties, the excited state structure is in some cases explained by the emergence of Baird aromaticity in the excited state[29], but there are also examples where the excited state is located to the arene fragment[39]. Therefore, we performed quantum chemical calculations of the ring inversion processes of $^{Th4}COT_{Saddle}$ and $^{Th6}CDH_{Screw}$ in the $S_0$, $S_1$, and $T_1$ states. The calculated transition state structures of $^{Th4}COT_{Saddle}$ in the $S_0$, $S_1$, and $T_1$ states are quasi-planar, as indicated by the small bent angles ($\theta$) of the COT core ($\theta = 6.2$, 13.8, and 3.7° for the $S_0$, $S_1$, and $T_1$ states, respectively; Fig. 4a). These geometries enable efficient $p_\pi$-orbital overlap, which is in line with observed $8\pi$-electron conjugation,

by means of molecular orbitals and spin densities, in the COT core (Supplementary Figs. 26 and 31). The alternation of C–C bond lengths in the COT core at the transition state in the $S_0$ state and equalization of those in the photoexcited ($S_1/T_1$) states demonstrate the antiaromatic and aromatic natures of the transition states in the $S_0$ state and photoexcited ($S_1/T_1$) states, respectively (Supplementary Fig. 40). Magnetic aromaticity indices[40], such as anisotropy of the induced current density (ACID)[40–42] and nucleus-independent chemical shift (NICS)[43–46], are commonly utilized to examine tentatively aromatic molecules. For the transition state structures in the $S_0$ and $T_1$ states, paratropic (counterclockwise, antiaromatic) and diatropic (clockwise, aromatic) ring currents, respectively, were observed in the ACID plots (Supplementary Figs. 34 and 36). The NICS$_{zz}$ scans, orthogonal to the central COT ring (Fig. 5a), also showed characteristic positive and negative minima for the transition states in the $S_0$ and $T_1$ states, respectively (Fig. 5c). On the other hand, the magnetic indices of aromaticity for the $S_1$ state are harder to assess because ACID is not available and NICS is not well-established (Supplementary Discussion). The NICS$_{zz}$ scan, which was obtained with CASSCF(8in8)/ 6–31 + + G(d,p) using Dalton 2016.0 (Supplementary Methods) and should be taken as qualitative rather than quantitative, showed the characteristic aromatic minima at the transition state in the $S_1$ state (Supplementary Fig. 39). These tentative findings are consistent with those of Solà and coworkers[14] who concluded that the corresponding state for the parent COT is highly aromatic according to electronic indices. These calculations indicate that the COT ring of $^{Th4}COT_{Saddle}$ exhibits aromatic nature in the transition state in the $T_1$ state and probably also in the $S_1$ state, while in the $S_0$ state it is antiaromatic.

Interestingly, 16π-electron circuits are formed due to conjugation of the COT core and two adjacent fused thiophene rings in the minimum energy structures of the photoexcited $^{Th4}COT_{Saddle}$. Although the minimum energy structures in the $S_1$ and $T_1$ states take a tub-shaped conformation ($\theta = 29.2°$ for $S_1$ and 28.7° for $T_1$ states; Fig. 4a), which is shallower than that in the $S_0$ state ($\theta = 40.3°$), they both have a bond-length equalized COT core. The ACID plot for the minimum energy structure in the $T_1$ state showed clear 16π-electron diatropic ring currents, which comprise the central COT ring and two adjacent thiophene rings (Supplementary Fig. 35). Such 16π-electron diatropic ring currents were not observed in the minimum energy structure in the $S_0$ state (Supplementary Fig. 33). The NICS(1)$_{iso}$ values were also consistent with the formation of a 16π conjugation pathway since both the COT ring (–8.9 ppm) and the thiophene rings (–7.3/–4.4 ppm above and below the ring, respectively) have negative NICS values. This is in sharp contrast with the NICS (1)$_{iso}$ values in the $S_0$ state: 0.0 ppm for COT and –6.8/–7.6 ppm (above/below) for the surrounding thiophene rings (Supplementary Table 7). These observations indicate some aromatic character due to 16π-electron circuits in the minimum energy structure of the photoexcited $^{Th4}COT_{Saddle}$. This highlights that a careful analysis that considers aromatic circuits from several rings is needed also for polycyclic systems in the excited state, similar as in the ground state[47, 48].

So why is the tub conformation preferred, even in the excited state? Calculations were performed on the model compound $^{Th2}COT_{Saddle}$, where two diagonally placed thiophene rings are removed to reduce steric repulsions. The optimization demonstrated that $^{Th2}COT_{Saddle}$ in the photoexcited ($S_1$/$T_1$) states prefers to be planar, which is in sharp contrast with the photoexcited $^{Th4}COT_{Saddle}$ (Supplementary Fig. 30). In addition, the activation enthalpy for ring inversion in the $S_0$ state is lowered to 6.3 kcal mol$^{-1}$ for $^{Th2}COT_{Saddle}$ from a calculated value of 29.9 kcal mol$^{-1}$ for $^{Th4}COT_{Saddle}$, indicating the effect of reduced steric strain. According to these model calculations, the shallow tub-shaped minimum energy structures of photoexcited $^{Th4}COT_{Saddle}$ represent a balance between maximizing the aromatic conjugation and minimizing the steric strain. Consequently, although a planar structure is favoured if only electronic effects are important, $^{Th4}COT_{Saddle}$ prefers a bent conformation in the excited state due to steric repulsion between neighboring thiophene rings.

For highly twisted $^{Th6}CDH_{Screw}$, DFT calculations showed multiple dihedral angles of ~90° in the [12]annulene core, indicating inhibition of full conjugation along the central core at the inversion transition states in the $S_0$, $S_1$, and $T_1$ states (Fig. 4b, Supplementary Fig. 44). This is in line with the observed spin density map at the transition state, showing that triplet excitation is localized to only one part of the annulene core (Supplementary Fig. 32). The C–C bond-length profile in the core at the transition states in the $S_0$, $S_1$, and $T_1$ states exhibits a clear alternating feature (Supplementary Fig. 41). These observations indicate the non-aromatic nature of the [12]annulene core of $^{Th6}CDH_{Screw}$ in the transition state. Because of the highly strained conformations of $^{Th6}CDH_{Screw}$, we can only utilize NICS calculations for the transition state structures as a magnetic index of aromaticity. As shown in Fig. 5d, the near-zero values of the NICS$_{zz}$ scans orthogonal to the central rings (Fig. 5b) suggest the non-aromatic nature of $^{Th6}CDH_{Screw}$. For the minimum energy structures, a non-aromatic nature is also expected for the $S_0$ and $T_1$ states because similar bond-length alternation and dihedral angle profiles were observed (Supplementary Figs. 43 and 45). The only exception is the minimum energy structure of the $S_1$ state, where bond-length equalization and small dihedral angles were observed, indicating the emergence of aromaticity. This

observation is consistent with the higher calculated activation enthalpy for the $S_1$ state (31.3 kcal mol$^{-1}$) compared to the $S_0$ (25.2 kcal mol$^{-1}$) and $T_1$ (23.0 kcal mol$^{-1}$) states, which could be attributed to the aromatic stabilization of the $S_1$ minimum that is lost at the transition state.

## Discussion

In 1972, Baird theoretically predicted that planar [4n]annulenes, which are energetically unfavorable in the electronic ground state according to Hückel's rule, become favoured upon photoexcitation. Using a particular [4n]annulene ($^{Th4}COT_{Saddle}$), which undergoes photo-accelerated ring inversion through its planar transition state (Fig. 1c), we succeeded in unambiguous experimental substantiation of Baird's rule from an energetic viewpoint. Additional important support was provided by the lack of photochemical acceleration in the ring inversion of a non-planarizable [4n]annulene ($^{Th6}CDH_{Screw}$; Fig. 1d). The energetic impact of Baird aromaticity (21–22 kcal mol$^{-1}$), determined by the analysis of the ring inversion kinetics of $^{Th4}COT_{Saddle}$, is noteworthy considering the stabilization energy of benzene of 28.8 kcal mol$^{-1}$, estimated based on experimental heats of formation[49]. Our study will help tailoring the potential energy surfaces of cyclic π-conjugated hydrocarbons with 4n π-electrons in their photoexcited states. Such compounds, if planar in the excited states, contribute as a new family of aromatic motifs to the progress of an exciting but much less explored area of organic photochemistry and related materials science.

## Methods

**Studies on the ring inversion events of $^{Th4}COT_{Saddle}$ and $^{Th6}CDH_{Screw}$.**
$^{Th4}COT_{Saddle}$ was subjected to chiral HPLC using hexane/CH$_2$Cl$_2$ (95/5 v/v) as eluent on a chiral DAICEL CHIRALPAK IF column (Supplementary Methods), and two well-separated enantiomer fractions were collected and evaporated to dryness at 25 °C. To the residue from the former or latter fraction was slowly added methylcyclohexane (3.0 mL), deaerated beforehand by Ar bubbling for 1 h, at 25 °C. The resulting solution was transferred to a quartz cuvette under Ar at 25 °C using a cannula to avoid contact with air and was utilized for the kinetic studies of the ring inversion of $^{Th4}COT_{Saddle}$ at 20 °C. As shown in Fig. 3c, plots of the CD intensity changes at 260 nm versus time gave the decay profiles in the $S_0$ (blue), $S_1$ (orange; excitation at 365 nm), and $T_1$ (green; excitation at 420 nm with fluorenone) states. $^{Th6}CDH_{Screw}$ is more subject to thermal ring inversion than $^{Th4}COT_{Saddle}$. Chiral HPLC separation of its enantiomers was conducted at 0 °C using hexane/EtOH (100/0.1 v/v) as eluent on a chiral DAICEL CHIRALPAK IA column (Supplementary Methods). To prevent thermal racemization, the collected enantiomer fractions were evaporated at –40 °C, and the residues were dissolved in chilled methylcyclohexane at –40 °C. At –20 °C under otherwise identical conditions, the resulting solutions were subjected to kinetic studies of the ring inversion of $^{Th6}CDH_{Screw}$ (Fig. 3d), and the CD intensity changes at 280 nm were plotted versus time. The kinetic analysis was performed according to the method described in the Supplementary Discussion.

**Quantum chemical calculations.** The molecular geometries in the $S_0$ and $T_1$ states were optimized using B3LYP[50] augmented by D3(BJ) dispersion[51] corrections and the 6–31G(d) basis set using Gaussian 09 Revision E.01[52] (for full reference, see Supplementary References). Stationary points, including the minimum energy and transition state structures, were confirmed by frequency calculations. For the transition state structures, IRC analysis was further carried out. Single-point electronic energies were calculated using the 6–311+G(d,p) basis set and used together with the corrections to the enthalpy taken from B3LYP-D3(BJ)/6–31G(d). For the $S_1$ state, we used TD-B3LYP/6–31+G(d,p) and both Gaussian 09 Revision E.01[52] and Gaussian 16 Revision A.03[53] (for full reference, see Supplementary References). The use of TD-DFT for the $S_1$ surface was validated for $^{Th4}COT_{Saddle}$ by comparison with ab initio methods (Supplementary Methods). NICS scans were calculated with GIAO-B3LYP/6–311+G(d,p) using the Aroma 1.0 package[43] (for full reference, see Supplementary References). ACID plots were calculated with CSGT-B3LYP/6–311+G(d,p).

**Data availability**. All relevant data are included in full within this paper and in the Supplementary Information.

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

## Acknowledgements

We thank Prof. Jun-ichi Aihara from Department of Chemistry, Shizuoka University for valuable discussions. We also thank Prof. Masahiro Yamanaka from Department of Chemistry, Rikkyo University for valuable discussions. K.J. and H.O. thank Prof. R. Herges for providing the AICD 2.0.0 program. This work was financially supported by a Grant-in-Aid for Specially Promoted Research (25000005) on "Physically Perturbed

Assembly for Tailoring High-Performance Soft Materials with Controlled Macroscopic Structural Anisotropy" for T.A. and Y.I.; M.U. thanks JSPS for a Young Scientist Fellowship and the Program for Leading Graduate Schools (MERIT). The work at Yonsei University was supported by Samsung Science and Technology Foundation under Project Number SSTF-BA1402-10. H.O. and K.J. acknowledge the Swedish Research Council (project grant 2015-04538) for support. Financial supports by Grant-in-Aids for Scientific Research, Challenging Exploratory Research, Promotion of Joint International Research (Fostering Joint International Research), and on Innovative Areas "Photosynergetics" (Grant Numbers JP15H03779, JP15K13642, JP16KK0111, and JP17H05261) from JSPS for T.M. is greatly acknowledged. Part of this work was conducted at the Advanced Characterization Nanotechnology Platform of the University of Tokyo, supported by the Ministry of Education, Culture, Sports, Science and Technology (MEXT), Japan. The Swedish National Infrastructure for Computation (SNIC) through NSC, Linköping, and HPC2N, Umeå, is acknowledged for computer time. A portion of the computations was performed using Research Center for Computational Science, Okazaki, Japan.

## Author contributions

M.U., T.A., and Y.I. initiated the work on the ring inversion kinetics of thiophene-fused annulenes. K.J. and H.O. proposed the incorporation of Baird aromaticity. M.U. and Y.I. performed the experiments and analyzed the data. K.J. performed the calculations, and, together with H.O., analyzed the data. Y.M.S. performed the transient absorption spectroscopies, and, together with D.K., analyzed the data. T.M. provided the theoretical basis for analyzing the CD intensity decay profiles to obtain the energetics of the system. M.U., K.J., H.O., T.A., and Y.I. wrote the paper. All authors discussed the results and commented on the manuscript.

## Additional information

**Competing interests:** The authors declare no competing financial interests.

