## [Peer Review File · Nature Communications]

Reviewers' comments:

Reviewer #1 (Remarks to the Author):

The authors report on two nonplanar thiophene-fused chiral [4n]annulenes, namely Th4 COT Saddle and Th6 CDH Screw obtained as by-products during the synthesis of oligothiophenes by a modified Ullmann coupling reaction.

Single-crystal X-ray crystallographic analysis revealed that these by-products are nonplanar and conformationally chiral. These compounds are an unprecedented pair of molecules that is suitable for energetic quantification of excited-state aromaticity. Their enantiomers were separable chromatographically but racemised thermally, enabling investigation of the ring inversion kinetics.

In contrast to Th6 CDH Screw, which inverts through a nonplanar transition state, the inversion of Th4 COT Saddle, progressing through a planar transition state, was remarkably accelerated upon photoexcitation. These findings are important since, as the authors state, the kinetic studies both in the S₀ state and photoexcited (S₁/T₁) state enable them to support Baird's rule from an energetic viewpoint. This conclusion is novel and will be of interest to the scientific community in the wide field of aromaticity.

The experiments and theoretical studies have been appropriately conducted and described and the work is also well supported by the literature. For these reasons, publication of this article in Nature Commun. with minor editorial revision is recommended.

Reviewer #2 (Remarks to the Author):

I suggest to accept for publication after minor changes.

The paper describes the preparation of cyclooctatetraene (**Th⁴COT**) and cyclohexadecahexaene (**Th⁶CDH**) derivatives, their X-ray-determined structures, the kinetics of their racemization and related computational studies. The main conclusion is that Baird aromaticity stabilizes the transition state for racemization of the **Th⁴COT** while not so in the racemization of the **Th⁶CDH**.

While the work is interesting and certainly deserve publication, there are some open questions that prevent me from recommendation to publish, "as is".

If I correctly understood the mechanism for photo-assisted racemization for the COT derivative, the first step is absorption of a photon, which leads to the S₁ and T₁ states. These states are still not planar (although "more planar" than the S₀ state). From there, there is racemization via an (almost) planar TS and finally returning to the S₀ state. If this is indeed so, Figure 1c is an incorrect picture. The red line should start at higher energy. If the absorption is at 365 nm, this is equivalent to ca. 78.4 kcal mol⁻¹, which should take the red curve in Figure 1c way up. Moreover, it should be emphasized that the S₁ and T₁ TSs are much higher in energy (by ca. 57 kcal mol⁻¹) than the S₀ TS. Thus, the double-head arrow in Figure 1c is misleading. Actually, the "red" TS is far higher in energy than the "black" TS. Please note that Baird aromaticity may be still operative in the red TS, but this TS is much less stable from the S₀ TS without any kind of aromaticity stabilization.

Can it be that the planar S₀ TS is destabilized by antiaromaticity? If so, can the authors estimate the energy of this destabilization and the Baird-aromatic stabilization energy, the sum of which is obviously 21-22 kcal mol⁻¹?

The planar **Th⁴COT** system can be thought about as a planar conjugated 24 π electrons (analogous to what the authors report, page 9, 2ed paragraph). Are the authors sure that this is not the case in the photo excited TSs? At least the ACID plots hint that this may be the case.

Finally, some technical remarks:

The manuscript should be proofread. For example, page 5 6th line from the bottom, last word – “deaerated” should probably “deuterated”?

As a user of Aroma, I know that it should be cited in a specific way. The authors should cite it properly.

Reviewer #3 (Remarks to the Author):

This paper is a thorough synthetic, spectroscopic and computational elucidation of an unusual photo-physical isomerization driven in part by “Baird’s Rule” of excited state aromaticity. A series of thienyl macrocycles were prepared, and it was noted that some of them had large barriers in the ground state for isomerization, whereas irradiated samples of a tetra-thiophene cycle (saddle conformer) had quick racemization. The authors identified the possibility for transition state stabilization in the case of excited molecules that, in a planar conformation, would exhibit 8 pi electron cycles that under the idea of Baird would have aromatic properties. This excited state property was shown to be operative for the singlet (direct excitation) and the triplet (through fluorenone sensitization) states. One example, 8 pi containing, had a planar transition state for the inversion, where another, 12 pi annulene, did not have planar transition state and seemed invariant to the influence of photoexcitation. (8 pi was not thermally convertible, while the 12 pi was) Although the idea of this Baird influence has been known previously for other 4n annulenes with the arene fusion groups, for the first time, the actual thermodynamic properties were calculated experimentally. The activation barriers were calculated based on the temperature dependence studies, as monitored by CD spectra over time to show the loss of chirality/racemization. This analysis presented a first experimental estimate of the energetic contributions of Baird stabilization in the excited states. Overall, I found this to be a convincing and well-researched initial study that is another example of the Baird rule for organic systems as has been championed by Ottosson in recent years. I would recommend publication after the authors attend to the following points:

Why were the data for Figure 3c/d and related presented as $\ln(CD_t/CD_0)$ that is, why the natural log?

In the statement: “The ring inversion of Th4COTSaddle was remarkably facilitated ($t_{1/2} = 2.1$ h) by the photoexcitation of fluorenone (Fig. 3c, green dots). Without photoirradiation, fluorenone itself did not facilitate the ring inversion (Fig. 3c, green dots).”
It seems they mentioned figure 3c green dots twice?

In Figure 3d, why is the scatter much more severe than in 3c?

Although the data and explanations do seem consistent, I think there needs to be some better explanation of the last aspect of the paper, regarding extensive aromatic visions (16 pi through two thiophenes and the annulene). What might help to clarify is if the authors can put some artificial bold arrows over the ACID maps to show the global direction of current mentioned.

Finally, I would like to see some discussion on two aspects: 1) alternative hypotheses that do not invoke Baird ideas, annulene planarity etc. I am not greatly familiar with the intricacies of these isomerizations, but I am well aware of many groups intensely studying the stilbene dynamics, where there are many other very subtle motions that collectively lead to isomerization cis-trans etc. (“hula twists” etc come to mind instantly). Are there other conformer twistings in these molecules that lead to the inversion without formal planarity in the COT ring? If so, how much more energetically costly are they vs the Baird idea here?

And 2) discussion about performing similar thermodynamic measurements on simpler annulenes (e.g. benzo oxepines referred in the text). It seems very likely that the smaller systems have

barriers to inversion that are too low to be able to isolate or observe isomers in any meaningful fashion under obtainable experimental conditions. If so, this should be noted, as it provides a novelty to this present work.

Reviewer #4 (Remarks to the Author):

This manuscript by Itoh and co-workers contains data for three crystal structures, denoted Th4-COT-Saddle, Th6-CDH-Screw [Redacted]. In general, the use of these structural models to support the conclusions of the manuscript is appropriate based on the data and refinement quality, and the structure refinements appear to have been carried out to a high standard. However, I would request the authors to attend to a couple of issues prior to this manuscript being accepted for publication in Nature Communications.

1) The Saddle and Screw structures are of good quality, however the Saddle structure is missing the crystal dimensions data from the cif file (though this is present in the data table). I will also point out that in this structure, modelling the four methyl group hydrogen atoms with an AFIX 137 model (allowing rotation about the C-C bond) rather than using AFIX 33 reduces the R1(I>2sigma) value by approximately 1% and accounts for much of the residual electron density in the vicinity of these sites - since there are no issues with data:parameter ratio with this structure I recommend this approach, as it will also slightly improve the precision of the rest of the structure. I would also suggest amending the ACTA card for this dataset to ACTA 52 or 50 rather than 55, as the completeness at 55 degrees is a little low.

[Redacted]

Point-to-Point Answers to Reviewers' Comments

Answers to Comments of Reviewer 1

The authors report on two nonplanar thiophene-fused chiral $[4n]$ annulenes, namely $^{Th4}COT_{Saddle}$ and $^{Th6}CDH_{Screw}$ obtained as by-products during the synthesis of oligothiophenes by a modified Ullmann coupling reaction. Single-crystal X-ray crystallographic analysis revealed that these by-products are nonplanar and conformationally chiral. These compounds are an unprecedented pair of molecules that is suitable for energetic quantification of excited-state aromaticity. Their enantiomers were separable chromatographically but racemised thermally, enabling investigation of the ring inversion kinetics. In contrast to $^{Th6}CDH_{Screw}$, which inverts through a nonplanar transition state, the inversion of $^{Th4}COT_{Saddle}$, progressing through a planar transition state, was remarkably accelerated upon photoexcitation. These findings are important since, as the authors state, the kinetic studies both in the S_0 state and photoexcited (S_1/T_1) state enable them to support Baird's rule from an energetic viewpoint. This conclusion is novel and will be of interest to the scientific community in the wide field of aromaticity. The experiments and theoretical studies have been appropriately conducted and described and the work is also well supported by the literature. For these reasons, publication of this article in *Nature Commun.* with minor editorial revision is recommended.

=> We highly appreciate this positive evaluation.

Answers to Comments of Reviewer 2

The paper describes the preparation of cyclooctatetraene (${}^{\text{Th}4}\text{COT}_{\text{Saddle}}$) and cyclohexadecahexaene (${}^{\text{Th}6}\text{CDH}_{\text{Screw}}$) derivatives, their X-ray-determined structures, the kinetics of their racemization and related computational studies. The main conclusion is that Baird aromaticity stabilizes the transition state for racemization of the ${}^{\text{Th}4}\text{COT}_{\text{Saddle}}$ while not so in the racemization of the ${}^{\text{Th}6}\text{CDH}_{\text{Screw}}$. While the work is interesting and certainly deserve publication, there are some open questions that prevent me from recommendation to publish, “as is”.

=> We highly appreciate this positive evaluation.

(1) If I correctly understood the mechanism for photo-assisted racemization for the COT derivative, the first step is absorption of a photon, which leads to the S_1 and T_1 states. These states are still not planar (although “more planar” than the S_0 state). From there, there is racemization via an (almost) planar TS and finally returning to the S_0 state. If this is indeed so, Figure 1c is an incorrect picture. The red line should start at higher energy. If the absorption is at 365 nm, this is equivalent to ca. 78.4 kcal mol⁻¹, which should take the red curve in Figure 1c way up. Moreover, it should be emphasized that the S_1 and T_1 TSs are much higher in energy (by ca. 57 kcal mol⁻¹) than the S_0 TS. Thus, the double-head arrow in Figure 1c is misleading. Actually, the “red” TS is far higher in energy than the “black” TS. Please note that Baird aromaticity may be still operative in the red TS, but this TS is much less stable from the S_0 TS without any kind of aromaticity stabilization.

=> According to this comment, we rearranged the energy profiles in Figs. 1c and 1d to prevent any misunderstanding of the readers.

(2) Can it be that the planar S_0 TS is destabilized by antiaromaticity? If so, can the authors estimate the energy of this destabilization and the Baird-aromatic stabilization energy, the sum of which is obviously 21–22 kcal mol⁻¹?

=> According to the previous research (*J. Chem. Theory Comput.* **8**, 1280–1287 (2012); Ref. 9 in the revised manuscript), the antiaromatic destabilization energy of D_{4h} COT is within the range of 1–3 kcal mol⁻¹. Considering that the planar TS of ${}^{\text{Th}4}\text{COT}_{\text{Saddle}}$ in the S_0 state has a COT core of bond-length alternating D_{4h} geometry (Fig. S40 in the revised Supplementary Information), the antiaromatic destabilization energy in this structure, if any, is most likely to be within the same range. An estimation of the (anti)aromatic (de)stabilization energy of the TS structures of ${}^{\text{Th}4}\text{COT}_{\text{Saddle}}$ requires a non-cyclically conjugated reference compound. However, this is not readily available in TSs and thus out of the focus in our study.

(3) The planar ${}^{\text{Th}4}\text{COT}_{\text{Saddle}}$ system can be thought about as a planar conjugated 24π electrons (analogous to what the authors report, page 9, 2nd paragraph). Are the authors sure that

this is not the case in the photo excited TSs? At least the ACID plots hint that this may be the case.

- => In the TS of the ring inversion of photoexcited ${}^{\text{Th4}}\text{COT}_{\text{Saddle}}$, there are several circuits available including a central 8π -electron circuit around the COT ring, a global 24π electron circuit around the perimeter of the molecule, as well as 16π -electron circuits encompassing the COT ring and two of the thiophene rings. Although the aromaticity of the TS is a sum of the effects from all these circuits, we are confident that the 8π -electron circuit is dominant. We support this by the fact that the diatropic ring current around the COT ring is stronger as evidenced by the longer arrows in the ACID plot (Fig. S36 in the revised Supplementary Information) and larger NICS(1) values as compared to the thiophene rings (-10 ppm vs. $-5-6$ ppm, Table S6 in the revised Supplementary Information). As previously shown, circuits with a larger surface area support higher ring current strengths even though they contribute less to the aromatic stabilization energy (*Phys. Chem. Chem. Phys.* **18**, 11847–11857 (2016); Ref. 48 in the revised manuscript). That the 8π -electron circuit has a higher current strength than the larger circuits despite it being smaller therefore shows that it has a much larger weight. The importance of considering aromatic circuits from several rings is emphasised in the text (pages 9–10, lines 27 and 1–2 in the revised manuscript).
- (4) Finally, some technical remarks: The manuscript should be proofread. For example, page 5 6th line from the bottom, last word – “deaerated” should probably “deuterated”?
- => The word “deaerated” is a word meaning that air was removed from the solution. In order to prevent any photochemical reaction caused by dissolved oxygen, we deaerated methylcyclohexane by purging with argon.
- (5) As a user of Aroma, I know that it should be cited in a specific way. The authors should cite it properly.
- => The reference to Aroma was indeed cited in the supplementary information (ref. S7 in previous Supplementary Information). In order to show this reference more clearly, we added the citation in the main reference (ref. 43 in the revised manuscript).

Answers to Comments of Reviewer 3

This paper is a thorough synthetic, spectroscopic and computational elucidation of an unusual photo-physical isomerization driven in part by “Baird’s Rule” of excited state aromaticity. A series of thienyl macrocycles were prepared, and it was noted that some of them had large barriers in the ground state for isomerization, whereas irradiated samples of a tetra-thiophene cycle (saddle conformer) had quick racemization. The authors identified the possibility for transition state stabilization in the case of excited molecules that, in a planar conformation, would exhibit 8π electron cycles that under the idea of Baird would have aromatic properties. This excited state property was shown to be operative for the singlet (direct excitation) and the triplet (through fluorenone sensitization) states. One example, 8π containing, had a planar transition state for the inversion, where another, 12π annulene, did not have planar transition state and seemed invariant to the influence of photoexcitation. (8π was not thermally convertible, while the 12π was) Although the idea of this Baird influence has been known previously for other $4n$ annulenes with the arene fusion groups, for the first time, the actual thermodynamic properties were calculated experimentally. The activation barriers were calculated based on the temperature dependence studies, as monitored by CD spectra over time to show the loss of chirality/racemization. This analysis presented a first experimental estimate of the energetic contributions of Baird stabilization in the excited states. Overall, I found this to be a convincing and well-researched initial study that is another example of the Baird rule for organic systems as has been championed by Ottosson in recent years. I would recommend publication after the authors attend to the following points:

=> We highly appreciate this positive evaluation.

(1) Why were the data for Figure 3c/d and related presented as $\ln(CD_t/CD_0)$ that is, why the natural log?

=> In order to utilize the integrated rate equation for first-order reaction ($\ln([A]/[A]_0) = -kt$), we employed natural log.

(2) In the statement: “The ring inversion of $^{Th4}COT_{Saddle}$ was remarkably facilitated ($t_{1/2} = 2.1$ h) by the photoexcitation of fluorenone (Fig. 3c, green dots). Without photoirradiation, fluorenone itself did not facilitate the ring inversion (Fig. 3c, green dots).” It seems they mentioned figure 3c green dots twice?

=> The former “green dots” refers to the red-striped zone in Fig. 3c and the latter one refers to the zone before and after the red-striped zone. In order to avoid confusion, we added this notion in the text (page 6, lines 13–15 in the revised manuscript).

(3) In Figure 3d, why is the scatter much more severe than in 3c?

=> Because $^{Th6}CDH_{Screw}$ racemises even at room temperature, it is technically difficult to avoid partial racemization while handling the optically resolved $^{Th6}CDH_{Screw}$ before

putting into the CD spectrometer. Therefore, the optical activity of the solution of $^{Th6}CDH_{Screw}$ used for the CD measurement is lower than that of an enantiomerically pure one. This is why the S/N ratio is relatively low compared to that of $^{Th4}COT_{Saddle}$. Note that the kinetic analysis of the racemization process is not dependent on the initial optical activity of the compound in solution.

- (4) Although the data and explanations do seem consistent, I think there needs to be some better explanation of the last aspect of the paper, regarding extensive aromatic visions (16π through two thiophenes and the annulene). What might help to clarify is if the authors can put some artificial bold arrows over the ACID maps to show the global direction of current mentioned.

=> According to this comment, we newly added supporting figures on the side of the ACID plots that help to understand the current direction on the rings.

Finally, I would like to see some discussion on two aspects:

- (5) Alternative hypotheses that do not invoke Baird ideas, annulene planarity etc: I am not greatly familiar with the intricacies of these isomerizations, but I am well aware of many groups intensely studying the stilbene dynamics, where there are many other very subtle motions that collectively lead to isomerization cis-trans etc. (“hula twists” etc come to mind instantly). Are there other conformer twistings in these molecules that lead to the inversion without formal planarity in the COT ring? If so, how much more energetically costly are they vs the Baird idea here?

=> A structural transition in COT which is in some way analogous to the “hula-twist” in double bond isomerization is called pseudorotation (*J. Am. Chem. Soc.* **112**, 239–253 (1990); Ref. S28 in the revised Supplementary Information) (Fig. CL1). It was significantly discussed around the 1990s for the ring inversion and bond shifting. The process of ring inversion from one tub conformer to its mirror image proceeds through an intermediate tub conformer via two pseudorotation transition states. For the parent COT, it is now widely accepted that the structural transition instead occurs through a planar TS in the ground state and not through pseudorotation (e.g. *Science* **272**, 1456–1459 (1996), *J. Comput. Chem.* **34**, 1393–1397 (2013); Ref. S29 and S30 in the revised Supplementary Information, respectively). However, in some COT derivatives with bulky substituents, pseudorotation pathways for structural transition from one tub conformation to the other one have been reported (e.g. *J. Am. Chem. Soc.* **135**, 14074–14077 (2013), *Org. Lett.* **19**, 2246–2249 (2017); Ref. S31 and S32 in the revised Supplementary Information). Also for the excited state of parent COT, the lowest-energy conical intersection of COT reachable from the S_1 state (12 kcal mol⁻¹ above the planar D_{8h} minimum that could also relax to either tub conformer) has a structure similar to a pseudorotation TS (*J. Am. Chem. Soc.* **124**, 13770–13789 (2002); Ref. 31 in the revised manuscript).

Figure CL1 | Pseudorotation of COT. Schematic illustrations of the process of pseudorotation of COT from one tub conformer (Tub-1) to the other (Tub-2). The coloured arrows represent the motion of the corresponding coloured carbon atoms during the pseudorotation.

Figure CL2 | Hypothetical energy diagram for pseudorotation of $\text{Th}^4\text{COT}_{\text{Saddle}}$. Schematic illustrations of a hypothetical energy diagram for pseudorotation process of $\text{Th}^4\text{COT}_{\text{Saddle}}$ from one saddle enantiomer (Tub-1) to the mirror image (Tub-2). The intermediate tub conformer of $\text{Th}^4\text{COT}_{\text{Saddle}}$ is very disfavoured due to steric repulsion of fused thiophene rings.

=> For our system, the ring inversion process for $\text{Th}^4\text{COT}_{\text{Saddle}}$ does not go through pseudorotation as the intermediate tub conformation would be destabilized due to coplanar thiophene rings (Fig. CL2). We have not been able to optimize any such intermediate by calculation. However, we think that the inversion of $\text{Th}^4\text{COT}_{\text{Saddle}}$ borrows some of the features of the pseudorotation pathway as evidenced by the not perfectly planar conformations of the TSs in S_0 , T_1 and S_1 states (Fig. 4a). A completely planar conformation (D_{2h} symmetry) is a higher order saddle point in all electronic states.

Additionally, we have also been able to optimize several other saddle points that all lie within 1 kcal mol⁻¹ that show that region around the TS is flat with significant structural flexibility (Fig. CL3). The difference from the ring inversion TS of COT itself is probably caused by the bulky thiophene groups. After investigating and rejecting the pseudorotation mechanism we are confident that the mechanism we present is the correct one. Additionally, the TS conformation is still nearly planar and, according to our analysis using BLA (Fig. S40 in the revised Supplementary Information), NICS (Fig. 5c), ACID (Figs. S34 and S36 in the revised Supplementary Information), as mentioned in the text, those TS structures, though not perfectly planar, can still maintain its cyclic conjugation.

=> We have added the above discussion in the revised Supplementary Information.

S₀ state

Symmetry	Relative electronic energy (kcal mol ⁻¹)	No. of Imaginary frequencies
C _i	0.00	1
D _{2h}	0.82	3

a) T₁ state

Symmetry	Relative electronic energy (kcal mol ⁻¹)	No. of imaginary frequencies
C _s	0.00	1
C _{2v}	0.09	3
C _{2h}	0.09	2
D _{2h}	0.46	5

b) S₁ state

Symmetry	Relative electronic energy (kcal mol ⁻¹)	No. of imaginary frequencies
C _s	0.00	1
C _{2h}	0.30	2
D _{2h}	0.65	5

Figure CL3 | Optimized saddle points of different order near the planar geometry of ^{Th4}COT_{Saddle} at the S₀, T₁ and S₁ states. Figures below each table shows an overlay of the optimized saddle points for the corresponding electronic states.

(6) Discussion about performing similar thermodynamic measurements on simpler annulenes (e.g. benzo oxepines referred in the text): It seems very likely that the smaller systems have barriers to inversion that are too low to be able to isolate or observe isomers in any meaningful fashion under obtainable experimental conditions. If so, this should be noted, as it provides a novelty to this present work.

=> We totally agree with this comment. We also think that the sufficient activation barrier in the ring inversion process of ${}^{\text{Th}4}\text{COT}_{\text{Saddle}}$, ${}^{\text{Th}6}\text{CDH}_{\text{Screw}}$ enabled us to discuss the energetic impact of Baird aromaticity. The simpler annulenes in the previous studies cannot be employed because of low or negative activation barrier. This is described in page 4, lines 7–14 in the revised manuscript.

Answers to Comments of Reviewer 4

This manuscript by Itoh and co-workers contains data for three crystal structures, denoted $\text{Th}^4\text{COT}_{\text{Saddle}}$, $\text{Th}^6\text{CDH}_{\text{Screw}}$ [Redacted]. In general, the use of these structural models to support the conclusions of the manuscript is appropriate based on the data and refinement quality, and the structure refinements appear to have been carried out to a high standard. However, I would request the authors to attend to a couple of issues prior to this manuscript being accepted for publication in Nature Communications.

=> We highly appreciate this positive evaluation.

(1) The Saddle and Screw structures are of good quality, however the Saddle structure is missing the crystal dimensions data from the cif file (though this is present in the data table). I will also point out that in this structure, modelling the four methyl group hydrogen atoms with an AFIX 137 model (allowing rotation about the C-C bond) rather than using AFIX 33 reduces the $R1(I>2\sigma)$ value by approximately 1% and accounts for much of the residual electron density in the vicinity of these sites - since there are no issues with data:parameter ratio with this structure I recommend this approach, as it will also slightly improve the precision of the rest of the structure. I would also suggest amending the ACTA card for this dataset to ACTA 52 or 50 rather than 55, as the completeness at 55 degrees is a little low.

=> According to this comment, we refined the structure again for improving the data. Accordingly, the crystal data in Table S1 in the revised Supplementary Information and the corresponding cif file has been revised.

[Redacted]

REVIEWERS' COMMENTS:

Reviewer #2 (Remarks to the Author):

Corrections and explanations in the cover letter satisfy me. I recommend publication.

Reviewer #3 (Remarks to the Author):

The authors have addressed my points appropriately, and this paper is now fit for publication.

Reviewer #4 (Remarks to the Author):

The authors have revised their manuscript and crystallographic data appropriately based on the previous round of refereeing, and I am now satisfied that the submission is suitable for publication in Nature Communications in its current form.